# Effect of Probiotic Supplementation on Gut Microbiota in Patients with Major Depressive Disorders: A Systematic Review

**DOI:** 10.3390/nu15061351

**Published:** 2023-03-10

**Authors:** Qin Xiang Ng, Yu Liang Lim, Clyve Yu Leon Yaow, Wee Khoon Ng, Julian Thumboo, Tau Ming Liew

**Affiliations:** 1Health Services Research Unit, Singapore General Hospital, Singapore 169608, Singapore; 2MOH Holdings Pte Ltd., 1 Maritime Square, Singapore 099253, Singapore; 3Department of Gastroenterology and Hepatology, Tan Tock Seng Hospital, Singapore 308433, Singapore; 4NUS Yong Loo Lin School of Medicine, Singapore 117597, Singapore; 5Department of Rheumatology and Immunology, Singapore General Hospital, Singapore 169608, Singapore; 6SingHealth Duke-NUS Medicine Academic Clinical Programme, Duke-NUS Medical School, Singapore 169857, Singapore; 7Department of Psychiatry, Singapore General Hospital, Singapore 169608, Singapore; 8Saw Swee Hock School of Public Health, National University of Singapore, Singapore 117549, Singapore

**Keywords:** probiotics, gut microbiota, gut composition, diversity, clinical trial

## Abstract

There is accumulating evidence on the beneficial effects of probiotic supplementation for patients with depressive disorders. However, prior reviews on the topic have largely focused on clinical effectiveness with limited emphasis on the underlying mechanisms of action and effects of probiotics on gut microbiota. In accordance with PRISMA guidelines, a systematic literature search of Medline, EMBASE and the Cochrane Library using combinations of the key words, (“depress*” OR “MDD” OR “suicide”), (“probiotic” OR “Lactobacillus” OR “Bifidobacterium”) AND (“gut” OR “gut micr*” OR “microbiota”), as well as grey literature was performed. We found seven clinical trials involving patients with major depressive disorder (MDD). The small number of studies and heterogeneous sources of data precluded meta-analysis. Most trials (other than one open-label trial) had a low-to-moderate risk of bias, which was largely due to a lack of control for the effects of diet on gut microbiota. Probiotic supplementation yielded only modest effects on depressive symptoms and there were no consistent effects on gut microbiota diversity, and in most instances, no significant alterations in gut microbiota composition were observed after four to eight weeks of probiotic intervention. There is also a lack of systematic reporting on adverse events and no good longer-term data. Patients with MDD may require a longer time to show clinical improvement and the microbial host environment may also need longer than eight weeks to produce significant microbiota alterations. To advance this field, further larger-scale and longer-term studies are required.

## 1. Introduction

Burgeoning research has highlighted the existence of a bidirectional communication pathway between the gut and the brain (also referred to as the ‘gut–microbiota–brain axis’) and the primacy of gut microbiota in numerous disease states, including cardiovascular diseases [1] and psychiatric disorders [2,3]. In particular, the gut microbiota appears to be a potential modifiable target for novel therapies, as it is closely influenced by the foods (e.g., prebiotics, probiotics and synbiotics) [4] and medicines (e.g., antibiotics) we consume [5].

The gut microbiota consists of billions of diverse bacteria, viruses, protozoa, archaea and fungi, and it is postulated that there is a complex bidirectional communication between the gastrointestinal tract and neural pathways [6], and disturbances in the gut microbiota can have profound effects on the development of disease states. In major depressive disorder (MDD), which is an exceedingly common psychiatric disorder and associated with a significant burden of disease [7] and impairments in health-related quality of life [8], there is accumulating evidence suggesting that increased peripheral and central pro-inflammatory cytokines [9] and reduced gut-microbial diversity [10] underlie its pathogenesis. Some research suggests that probiotics may have an effect on the gut–brain axis, and that changes in the gut microbiome may be linked to mood disorders such as depression. Although probiotics show promising preclinical data for their potential anti-depressive effects, clinical trials have yielded heterogeneous effects [11,12], and prior reviews on the topic have hitherto focused on the clinical effectiveness and less on the underlying mechanisms of action [2,3].

Depression affects more than 250 million people worldwide [13]. MDD in particular is the leading cause of disability across the world, in terms of years lived with disability. MDD may be very resistant to treatment, is significantly debilitating and may even lead to suicide in affected individuals [7,8]. The burden of MDD extends well beyond the individual with MDD and can have far-reaching adverse effects on families, communities, and society as a whole. MDD sufferers also have an increased risk of developing medical conditions such as cardiovascular disease and diabetes, and may incur higher attendant healthcare costs [14]. Despite the global prevalence and disease burden of MDD, up to one-third of patients with MDD respond partially or fail to respond to current first-line therapies (such as antidepressant medications or psychotherapy) [15], highlighting the need for newer and more effective therapeutic strategies. Probiotics, which are live microorganisms that are similar to the beneficial microorganisms found in the human gut [16], have been studied for their potential role in treating depression. Probiotics can be found in fermented foods such as yoghurt, kefir and sauerkraut, and are also available over-the-counter as dietary supplements. Most commercial probiotic products contain *Bifidobacterium* spp. and *Lactobacillus* spp., in the range of 1 × 10^9^ colony forming units (CFU)/g [17,18]. Given that they are relatively safe and readily acceptable to patients compared to conventional anti-depressants, it is worth investigating the evidence base for these emerging therapies for patients with MDD.

Indeed, the current state of the art on probiotic therapy for depression is still evolving, with ongoing research aimed at better understanding the potential mechanisms of action and the optimal use of probiotics for this condition. In an attempt to unravel the underlying mechanisms and generate hypotheses for future investigations, this review aimed to specifically examine the effects of probiotic supplementation on the gut microbiota in patients with MDD.

## 2. Methods

The proposed review protocol was prospectively registered in PROSPERO (registration number CRD42023387500). A systematic literature search was conducted in accordance with the latest PRISMA guidelines [19] and performed in Medline, EMBASE and Cochrane Library databases. Combinations of the following key search terms, (“depress*” OR “MDD” OR “suicide”), (“probiotic” OR “Lactobacillus” OR “Bifidobacterium”) AND (“gut” OR “gut micr*” OR “microbiota”) were used, and the search period was defined as from database inception up until 1 December 2022. The full search strategy (with the exact search terms and operators) used can be found in the Appendix A. Attempts were made to search grey literature using Google.

The inclusion criteria for the review included (1) original studies published in English, (2) clinical trials, (3) involving patients with depression, (4) defined probiotic intervention and (5) documented changes in gut microbiota. Abstracts were screened using Covidence online software (Melbourne, VIC, Australia) by three independent researchers (Q.X.N., C.Y.L.Y. and Y.L.L.). Full texts were obtained for all articles of interest and their reference lists were manually searched to identify additional relevant papers. Subject content experts were also consulted to identify additional relevant articles. Full articles were assessed thoroughly for eligibility based on predefined inclusion and exclusion criteria. Conflicts were resolved by discussion and consensus with the senior authors.

The primary objective of the study was to investigate the effect of probiotic supplementation on gut microbiota. The secondary objective included a change in depressive symptoms, as rated by validated rating scales, e.g., Hamilton Depression Rating Scale (HAM-D) or Beck Depression Index (BDI), post intervention. Where possible, the results were pooled using a random effects meta-analysis, with standardized mean differences (SMD) calculated as the studies used different scales and the outcome measurements had different units across the trials reviewed. SMD expresses the size of the treatment effect in standard deviation (SD) units, rather than in the original units of the outcome measure [20]. Statistical analyses were performed in R 4.0.3 (R Foundation for Statistical Computing, Vienna, Austria).

Once a final set of eligible studies were identified, relevant data from the studies were extracted using a standardized data extraction form by two study investigators (C.Y.L.Y. and Y.L.L.) and cross-checked by a third (Q.X.N.) for accuracy.

The risk of bias of the studies was appraised using Version 2 of the Cochrane risk-of-bias tool for randomized trials (RoB 2) [21] via consensus of three study investigators (Q.X.N., C.Y.L.Y. and Y.L.L.). The tool examines several key domains, including the adequacy of sequence generation, allocation sequence concealment, blinding, incomplete outcome data, selective outcome reporting and the presence of other potential sources of bias inherent to the study under review [21].

## 3. Results

After a comprehensive literature search, a total of 1395 citations were identified. After removing duplicates and screening the records based on their titles and abstracts, 15 full-texts remained. Eight studies were excluded for the reasons mentioned in Figure 1, leaving seven studies [11,12,22,23,24,25,26] eligible for review (Figure 1). The key characteristics and salient findings of the studies are summarized in Table 1. Probiotic intervention in the studies reviewed was variable, with differences in the type of probiotics, dosage, and duration of follow-up. As the number of available studies were limited (<10) and had dissimilar designs and diverse sources of data, there was a limited scope for meta-analysis [27] and a narrative synthesis of the findings was performed instead.

Except for an open-label trial [23], the other randomized, controlled trials generally had a low-to-moderate risk of bias (full breakdown shown in Table 2). A significant issue in the studies reviewed was that they did not control for the effects of diet on gut microbiota.

In general, probiotic supplementation did not seem to significantly improve depressive symptoms compared to placebo [11,22,24,26] and answering the primary objective of our review, the majority of the studies failed to find significant alterations in gut microbiota composition [11,12,22,23,24,26].

In terms of the secondary outcome of our review, we confirmed that based on a meta-analysis of five studies (excluding the open trial by Chen et al. [23] and Reininghaus et al. [11] for potentially overlapping populations), the pooled SMD for change in depressive rating scales was −0.50 (95% CI: −1.13 to 0.14, τ^2^ = 0.16, I^2^ = 69%), which indicated that probiotic supplementation did not significantly improve depressive symptoms as compared to placebo (forest plot shown in Appendix A).

## 4. Discussion

Based on the available studies, probiotic supplementation appeared to have limited effects on depressive symptoms and gut microbiota in patients with MDD. MDD is a complex, heterogeneous illness, and with standard antidepressant treatment, it may take two to three months before the symptoms of MDD improve and even longer to achieve clinical remission [28]. The response to treatment can also vary greatly between individuals. Similarly, from a molecular perspective, the microbial host environment may need longer than eight weeks (which was the study duration in most instances) to show significant microbiota alterations [29]. With short-term probiotic intervention, the shift in the gut microbiota may also be transient and temporary.

Probiotics research is a rapidly growing field that cuts across biology, translational medicine, epidemiology and bioinformatics, and it aims to understand the beneficial effects of probiotics on human health and disease states. It is thought that gene–environment and gut–microbiota–brain interactions lead to a concatenation of events that influence brain function and mood disorders. Earlier studies have found the gut flora of depressed patients to have lower levels of certain beneficial bacteria, such as *Lactobacillus* and *Bifidobacterium*, and a higher proportion of pathogenic Gram-negative bacteria, such as *Enterobacteriaceae* [30,31]. In the available studies, alpha diversity was typically assessed using richness and the Shannon diversity index, and the authors also studied beta diversity, or the dissimilarity of two communities. In most studies, however, the relative abundance of the main phyla identified (Bacteroidetes, Firmicutes, Actinobacteria, Proteobacteria, Fusobacteria, Synergistetes, Verrucomicrobia, and Euryarchaeota) did not differ significantly despite probiotic supplementation [11,12,22,23,24,26]. The richness of microbial diversity referred to the number of genera detected within each fecal sample.

In the PROVIT trials [11,24], there was no significant difference in terms of psychiatric symptom rating scales between the probiotic intervention and placebo control group, but a relatively increased beta diversity and abundance of *Ruminococcus gauvreauii* and *Coprococcus 3* was observed after four weeks of probiotic supplementation. *Ruminococcus* and *Coprococcus* are common butyrate-producing bacteria in the human gut; it is thought that these short-chain fatty acids are essential for intestinal barrier integrity, may exhibit anti-inflammatory activity and contribute to the up-regulation of brain-derived neurotrophic factor (BDNF) when they enter the bloodstream [32,33]. In perhaps the most positive trial by Tian et al. [25], which found significant improvements in depressive symptoms, this was accompanied by a relative abundance of *Desulfovibrio* and *Faecalibaculum* after four weeks of probiotic supplementation. This agrees with earlier studies that found patients with MDD to have a relative abundance of *Enterobacteriaceae* and *Alistipes* but lowered levels of beneficial *Faecalibacterium* [30]. However, the findings on the gut microbiota in patients with MDD have been inconsistent [34]. This is further complicated by the fact that there is no consensus on the ‘ideal’ gut microbiota composition for optimal physical functioning and mental well-being [33,34].

Corroborating our findings, an earlier systematic review of randomized, controlled trials in healthy participants [35] found largely similar results to our current study. Kristensen et al. (2016) reviewed seven clinical trials investigating alterations in the microbial composition of healthy human fecal samples via high-throughput molecular approaches, and only one study reported significant changes in terms of beta diversity (compositional dissimilarity) after probiotic supplementation, compared to the placebo group [35]. Similar to our review on patients with MDD, the authors concluded a lack of consistent evidence on the positive effect of probiotics on fecal microbiota composition in healthy adults. This is perhaps unsurprising as the Human Microbiome project revealed the microbial taxa complexity in the human gut, with high inter-individual and day-to-day variability due to inheritance, diet, environmental and other factors [36]. Part of the source for heterogeneity in the studies probably also stems from the heterogeneous nature of the diagnosis of MDD and condition [37]. As discussed earlier, the pathogenesis of MDD is multifaceted and not yet fully understood, but it is likely to involve a complex interplay of genetic, environmental and neurobiological factors.

On balance, there is insufficient evidence in the current literature to support the short-term effects of probiotic supplementation on depression or the gut microbiota. Previous meta-analyses on the topic have supported the potential use of probiotics as an adjunctive therapy for depression, but noted that further research is needed to determine the optimal dose, duration and strain of probiotics for this condition [2,38,39]. Collectively, the meta-analyses emphasized the heterogeneity of the available studies, the overall small to moderate effect size and the importance of taking into account the variable individual differences in gut microbiota composition and the myriad host factors that may influence treatment response. It is evident that more research is needed to fully understand the mechanisms through which probiotics exert their effects. Subgroup analyses in the meta-analysis by Goh et al. (2019) also suggested that not all probiotics are the same [38]. Compared to single-strain probiotic products, multi-strain products had a significant effect on alleviating depressive symptoms, even though the combinations of species and strains of probiotics tested with were too varied to conclude specific effective probiotic strains [38]. Nonetheless, there remains a fundamental contention on whether probiotic treatments can successfully alter microbiota composition.

The gut microbiota is a complex and intricate ecosystem of microorganisms that inhabit the human gastrointestinal tract. Modern gut microbiota analyses provide data on the relative and not absolute abundance of particular bacterial species within the gut. Consequently, elevations in abundance do not need to be equal to the increase within the ecological milieu per se. Furthermore, the 16S rRNA sequencing method used in existing studies may not be sensitive enough to detect the microbiota alterations induced by probiotic administration, as compared to newer techniques such as shotgun metagenomics and RNA sequencing. These techniques are more sensitive, have greater resolution and provide a more comprehensive picture regarding the structure and function of host microbial communities [40]. Moreover, although it is widely believed that probiotics work by colonizing the gut and modulating the composition and activity of the gut microbiota, it is assumed but not proven that for a probiotic to confer therapeutic benefits, it has to significantly alter the gut microbiota of the host. Rather, the therapeutic benefits could be due to other mechanisms or accrued through metabolites produced by the probiotic strains as they pass through the intestine, and through intricate interactions with the host’s metabolism and immune system in vivo. Probiotics may influence the production and activity of neurotransmitters; some probiotic strains have been shown to regulate host serotonin biosynthesis, increasing the production of serotonin, which is a neurotransmitter that is involved in mood regulation [41]. Other probiotics have been found to affect the production of dopamine, norepinephrine and GABA neurotransmitters [42]. In a pilot study, probiotics were found to modulate gut microbiota gene expression in the absence of compositional changes, with potential anti-inflammatory effects [43]. This is another potential mechanism through which probiotics may affect metabolic function. In sum, probiotics may have beneficial effects on gut–brain communication, inflammation, and neurotransmitter signaling, which could potentially contribute to their antidepressant effects. Alas, the specific mechanisms of action are still not fully understood.

It is also worth acknowledging that the available studies only looked at the short-term effects of probiotic supplementation on symptom improvement and gut microbiota changes, and little is known about the long-term effects, especially for relapse prevention in a chronic illness such as MDD. MDD is a chronic and relapsing condition, with at least 50% of people experiencing recurrent episodes of depression after an initial episode [44]. Although the available studies do not indicate an increased risk of serious side effects, there is a lack of systematic reporting on adverse events and no good longer-term data are available [45]. There are anecdotal reports that probiotics may worsen outcomes [46], and in fecal microbiota transplant recipients for example, reports of gastrointestinal disturbances, e.g., diarrhea, abdominal pain and bloating are not uncommon [47], and this has important implications for short-term treatment and long-term management. More research is needed to fully understand the utility of probiotics for treating depression.

An additional consideration is the immune response to probiotics by the human host. Immune responses such as phagocytosis and xenophagy, and methods of programmed cell-death such as pyroptosis and necroptosis [48,49], render most invaders harmless and microbial infections self-limiting. Our immune system also contributes to nutrition acquisition by degrading human microbiota, pathogens and damaged body tissue cells, as it is able to utilize some of the metabolic products from these microorganisms as well as the infected gastric epithelial or somatic cells as a source of essential nutrients [50,51]. As the gut microbiome is composed of 150 times more genes than those found in the entire human genome, the human microbiome is an indispensable source of metabolites for the human body [52,53], and these may modulate the overall effect on health and disease states.

In terms of the limitations to our systematic review, first, the small number of studies (and sample sizes), heterogeneous sources of data and differing outcome measurements precluded the possibility of meta-analysis. For the secondary study outcome on the effect of probiotics on depressive symptoms, as there were only five studies included, it is difficult to draw any firm conclusions. Second, several of the included studies were from Asia and all the studies had a preponderance of female participants (>60 to 70%), which would give rise to gut microbiota variations. It is known that males and females have distinct gender-specific differences in terms of host bacterial genera [54]. Third, the majority of the studies reviewed did not control for the effects of diet on gut microbiota; besides pre/probiotics, dietary polyphenols, which are often indigestible, would also influence gut microbiota diversity and composition [55]. The medications used to treat MDD and behavioral conditions related to MDD (e.g., dietary habits, sedentary lifestyle, etc.) can also disturb the composition of the gut microbiota. The compositionality problem inherent to microbiome investigations is another factor affecting the results of microbiota analysis and may be largely responsible for the inconsistency of the results obtained in existing microbiota studies. To address some of these issues, there are several reporting guidelines for human microbiome research, which aim to reduce reporting heterogeneity and improve the strength, quality and transparency of reporting for studies in this interdisciplinary field. For example, the STORMS (STrengthening the Reporting Of Microbial Studies) checklist is a 17-item checklist tool developed to guide authors and reviewers in the reporting and evaluation of studies related to the human microbiome [56]. Microbial genome sequences (including the sequencing platform, quality control measures, and data analysis) should be duly reported. At a minimum, the STORMS checklist recommends that descriptive statistics (including the age and gender of the study population) and the main study outcomes and the results of any additional analyses should be detailed. This ensures that the description of the study is complete and organized. Fourth, the gut microbiome is also primarily studied using fecal bacterial communities as a surrogate. Fecal samples are broadly representative of colonic luminal bacteria; however, some communities of bacteria may be overlooked, which may also explain the difference seen between studies that utilized mucosal as opposed to fecal samples as a proxy [57]. Last but not least, further mechanistic studies are necessary to elucidate the effects of probiotics on depression, which are likely multifactorial and involve complex interactions between the host gut microbiota, immune system and central nervous system. The use of “multi-omics” technologies, as adapted from previous studies [58], might help shed light on these questions.

## 5. Conclusions

In conclusion, based on available clinical trials, probiotic supplementation yielded limited short-term effects on gut microbiota in patients with MDD, and produced only modest effects on depressive symptoms in the studies reviewed. There were variable colonization patterns observed but most studies failed to find significant alterations in gut microbiota composition after four to eight weeks of probiotic intervention, with the caveat being the small sample size and short treatment duration of present studies. To advance this field, further larger-scale and longer-term studies are required. Future studies could also consider alternative approaches to study gut microbiota, e.g., metagenomics and RNA sequencing, which may be more sensitive, have greater resolution and provide a more comprehensive picture regarding the structure and function of host microbial communities.

## Figures and Tables

**Figure 1 nutrients-15-01351-f001:**
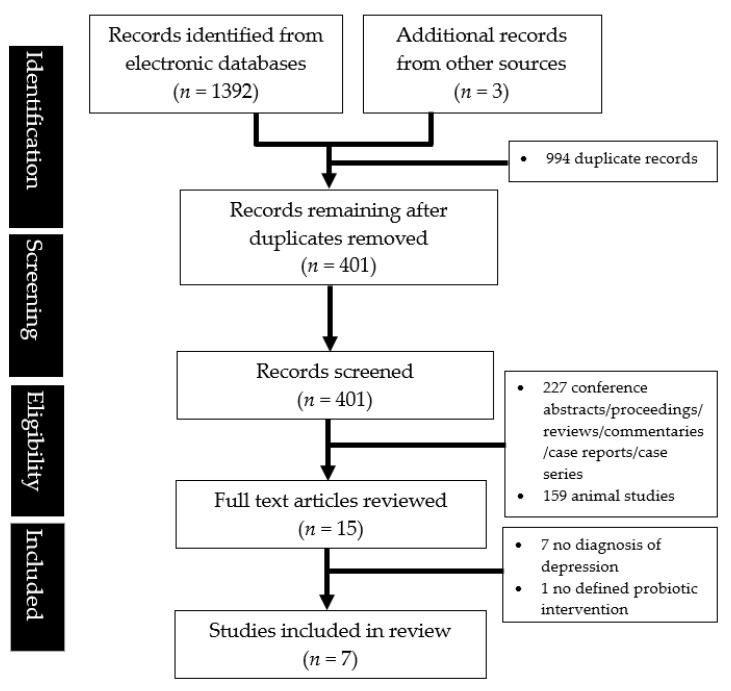
PRISMA flowchart showing the study abstraction process.

**Table 1 nutrients-15-01351-t001:** Relevant characteristics and findings of the studies reviewed (arranged alphabetically by the first author’s last name).

Author, Year	Country	Study Design	Study Population and Sample Size (N)	Intervention	Gut Microbiota Analysis	Key Findings
Chahwan et al., 2019 [22]	Australia	Triple-blinded parallel, placebo-controlled randomized trial	Mean age 36.65 (SD ± 11.75), *n* = 71, 61.8% female, patients with diagnosis of MDD and BDI-II score ≥ 12	A total of 2 g of freeze-dried probiotic powder mixture (*B. bifidum* W23, *B. lactis* W51, *Bifidobacterium lactis* W52, *L. acidophilus* W37, *L. brevis* W63, *L. casei* W56, *L. salivarius* W24, *L. lactis* W19 and *L. lactis* W58, total cell count 1 × 10^10^ CFU/day) twice daily	16S rRNA gene sequencing of bacterial DNA isolated from stool samples, Illumina MiSeq platform and QIIME 1.9.1 pipeline	Although patients with MDD did show significantly reduced cognitive reactivity (*p* = 0.04), the gut microbiota composition was similar in all groups.
Chen et al., 2021 [23]	Taiwan	Open-label trial	Mean age 39.4 (SD ± 12.0), *n* = 11, 72.7% female, patients with diagnosis of MDD and HAMD-17 scores ≥ 14	1 capsule (3 × 10^10^ CFU of *L. plantarum* PS128) daily	16S rRNA gene amplification sequences of bacterial DNA isolated from stool samples, Illumina MiSeq platform and processed using QIIME2 pipeline	Despite improvements in depressive and somatic symptoms, the composition of gut microbiota was not significantly altered after 8 weeks of probiotic supplementation.
Kreuzer et al., 2022 [24]	Austria	Double-blind, randomized, placebo-controlled trial	Mean age 44.63 (SD ± 15.12), *n* = 28, 75% female, patients with MDD, mean BDI-II 31.11 and mean HAMD 15.14	Probiotic drink containing ≥ 2.5 × 10^9^ CFU/g *B. bifidum* W23, *B. lactis* W51, *B. lactis* W52 and *L. acidophilus* W22, *L. casei* W56, *L. paracasei* W20, *L. plantarum* W62, *L. salivarius* W24, *L. lactis* W19. Additionally, it contained other ingredients including D-biotin (Vitamin B7), common horsetail, fish collagen, keratin, inulin, etc.	16S rRNA gene amplicon sequencing of bacterial DNA isolated from stool samples, using Illumina MiSeq and QIIME 1.9.1	No significant difference in terms of psychiatric rating scales between intervention and control group, but probiotic supplementation resulted in a higher relative abundance of *Coprococcus* 3 and *Ruminococcus grauvanii*, which corresponded to higher normalized concentrations of butyrate, alanine, valine, isoleucine, sarcosine, methylamine, and lysine amino acids.
Reininghaus et al., 2020 [11]	Austria	Double-blind, randomized, placebo-controlled trial	Mean age 43 (SD ± 14.31), *n* = 42, 71.4% female, patients with MDD, mean BDI-II 30.75 and mean HAMD 15.07	Probiotic drink containing ≥ 2.5 × 10^9^ CFU/g *B. bifidum* W23, *B. lactis* W51, *B. lactis* W52 and *L. acidophilus* W22, *L. casei* W56, *L. paracasei* W20, *L. plantarum* W62, *L. salivarius* W24, *L. lactis* W19. Additionally, it contained other ingredients including D-biotin (Vitamin B7), common horsetail, fish collagen, keratin, inulin, etc.	16S rRNA gene amplicon sequencing of bacterial DNA isolated from stool samples, using Illumina MiSeq and QIIME 1.9.1	No difference in terms of psychiatric rating scales between intervention and control group, but relatively increased beta diversity and abundance of *Ruminococcus gauvreauii* and *Coprococcus* 3 after 4 weeks of probiotic supplementation.
Schaub et al., 2022 [12]	Switzerland	Double-blind, randomized controlled trial	Mean age 39.43 (SD ± 11.45), *n* = 21, 67% female, patients with MDD and HAMD ≥ 7	Probiotic supplement containing *Streptococcus thermophilus* NCIMB 30438, *B. breve* NCIMB 30441, *B. lactis* NCIMB 30435, *B. infantis* NCIMB 30436, *L. acidophilus* NCIMB 30442, *L. plantarum* NCIMB 30437, *L. paracasei* NCIMB 30439, *L. helveticus*, daily dose contained 900 billion CFU/day	16S rRNA gene sequencing of bacterial DNA extracted from stool samples, using DADA2 pipeline	Significant improvement in depressive symptoms in intervention group (*p* < 0.05); alpha diversity measures showed no significant changes but relative abundance of *Lactobacillus* genera after 4 weeks of probiotic supplementation.
Tian et al., 2022 [25]	China	Double-blind, placebo-controlled, randomized trial	Mean age 51.32 (SD ± 16.11), *n* = 20, 70% female, patients with MDD and HAMD-24 score ≥ 14	Sachet of freeze dried *B. breve* CCFM1025 powder (total 10^10^ CFU) daily	16S rRNA gene amplicon sequencing of bacterial DNA extracted from stool samples, processed using QIIME2	Significant improvements in depressive symptoms, accompanied by relative abundance of *Desulfovibrio* and *Faecalibaculum* after 4 weeks of probiotic supplementation.
Zhang et al., 2021 [26]	China	Double-blind, placebo-controlled, randomized trial	Mean age 45.8 (SD ± 12.3), *n* = 38, 63.2% females, patients with MDD, HAMD-17 score ≥ 8 and diagnosis of constipation	100 mL of probiotic drink containing 10^8^ CFU/mL of *L. paracasei* strain Shirota	16S rRNA gene sequencing of bacterial DNA extracted from stool samples, using Illumina MiSeq platform and QIIME	Significant improvements in constipation and depressive symptoms albeit not statistically significant between groups; probiotic supplementation for 9 weeks did not alter beta diversity but slightly increased levels of *Adlercreutzia*, *Megasphaera* and *Veillonella* genera and decreased *Rikenellaceae_RC9_gut_group*, *Sutterella* and *Oscillibacter*.

Abbreviations: BDI-II, Beck Depression Index-Second Edition; HAMD, Hamilton Depression Rating Scale; MDD, major depressive disorder; QIIME, Quantitative Insights Into Microbial Ecology.

**Table 2 nutrients-15-01351-t002:** Risk of bias assessment for the studies reviewed.

Study, Year	Sequence Generation	Allocation Concealment	Blinding	Incomplete Results	Selective Reporting	Other Bias(es)
Chahwan et al., 2019 [22]						
Chen et al., 2021 [23]						
Kreuzer et al., 2022 [24]						
Reininghaus et al., 2020 [11]						
Schaub et al., 2022 [12]						
Tian et al., 2022 [25]						
Zhang et al., 2021 [26]						

Interpretation: ........ high, ........ unclear and ........ low risk of bias.

## Data Availability

The datasets generated during and/or analyzed during the current study are available from the corresponding author on reasonable request.

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
