# Peer review of "Effect of Probiotic Supplementation on Gut Microbiota in Patients with Major Depressive Disorders: A Systematic Review"

_nutrients, 2023, doi:10.3390/nu15061351_

Round 1

Reviewer 1 Report

This manuscript is thought-provoking, maintaining a questioning attitude towards academics and pointing out the problems of clinical trials of probiotics in the treatment of Major Depression Disorder. However, the authors should accept some revisions of the manuscript, particularly on the following points:

1.      Does this manuscript show that probiotics have no significant effect on patients with major depression? 

2.      Clinical trials of probiotics for severe depressive disorder are rare and clinical data are insufficient. Only 7 articles were screened out, and it was difficult to draw the exact conclusion from the 7 clinical trials.

3.      The purpose of using probiotics is to treat Major Depressive Disorder by changing the gut microbiota, and ultimately achieve the purpose of reducing the degree of depression in patients. However, is it possible that no significant changes are observed in the gut microbiota, but there is a significant reduction in the degree of depression in patients?

Author Response

Comment 1: Does this manuscript show that probiotics have no significant effect on patients with major depression? 

Reply 1: Thank you for the comment. We acknowledge in our study limitations that "For the secondary study outcome on the effect of probiotics on depressive symptoms, as there were only five studies included, it is difficult to draw any firm conclusions." In our discussion, we also qualify that "although it is widely believed that probiotics work by colonizing the gut and modulating the composition and activity of the gut microbiota, it is assumed but not proven that for a probiotic to confer therapeutic benefits, it has to significantly alter the gut microbiota of the host. Rather, the therapeutic benefits could be due to other mechanisms or accrued through metabolites produced by the probiotic strains as they pass through the intestine, and through intricate interactions with the host’s metabolism and immune system in vivo."

Comment 2: Clinical trials of probiotics for severe depressive disorder are rare and clinical data are insufficient. Only 7 articles were screened out, and it was difficult to draw the exact conclusion from the 7 clinical trials.

Reply 2: Thank you for the comment. We agree with the reviewer and have added this in our discussion of study limitations; "For the secondary study outcome on the effect of probiotics on depressive symptoms, as there were only five studies included, it is difficult to draw any firm conclusions."

Comment 3: The purpose of using probiotics is to treat Major Depressive Disorder by changing the gut microbiota, and ultimately achieve the purpose of reducing the degree of depression in patients. However, is it possible that no significant changes are observed in the gut microbiota, but there is a significant reduction in the degree of depression in patients?

Reply 3: Thank you for the comment. We agree with the reviewer and in our discussion section, we also qualify that "although it is widely believed that probiotics work by colonizing the gut and modulating the composition and activity of the gut microbiota, it is assumed but not proven that for a probiotic to confer therapeutic benefits, it has to significantly alter the gut microbiota of the host. Rather, the therapeutic benefits could be due to other mechanisms or accrued through metabolites produced by the probiotic strains as they pass through the intestine, and through intricate interactions with the host’s metabolism and immune system in vivo." We also discuss other potential therapeutic mechanisms, e.g. "Probiotics may influence the production and activity of neurotransmitters; some probiotic strains have been shown to regulate the host serotonin biosynthesis, increasing the production of serotonin, which is a neurotransmitter that is involved in mood regulation [41]. Other probiotics have been found to affect the production of dopamine, norepinephrine, and GABA neurotransmitters [42]. In a pilot study, probiotics were found to modulate gut microbiota gene expression in the absence of compositional changes, with potential anti-inflammatory effects [43]. This is another potential mechanism by which probiotics may affect metabolic function. In sum, probiotics may have beneficial effects on gut-brain communication, inflammation, and neurotransmitter signaling, which could potentially contribute to their antidepressant effects. Alas, the specific mechanisms of action are still not fully understood."

Reviewer 2 Report

This systematic review paper reviewed the beneficial effects of probiotic supplementation for patients with major depressive disorders (MDD) with emphasis on the underlying mechanisms of action and effects of probiotics on gut microbiota. The results show that, probiotic supplementation yielded limited effects on gut microbiota in patients with MDD, and produced only modest effects on depressive symptoms in the studies. There were variable colonization patterns observed but most studies failed to find significant alterations in gut microbiota composition after four to eight weeks of probiotic intervention. While the topic addressed in this systematic review is very interesting, several other factors may also be taken into consideration. As pointed out by the authors, the therapeutic benefits probiotic supplementation could be due to other mechanisms or accrued through metabolites produced by the probiotic strains as they pass through the intestine, and through intricate interactions with the host’s metabolism and immune system.

When we talk about gut microbiota, we need to take the human host immunity response into account [1-3]. Thanks to the immune responses like phagocytosis, xenophagy, and programmed cell-death like pyroptosis and necroptosis [4], most of microorganism infections are self-limiting (and become the so-called gut microbiota), and our immune system will use metabolic products of these microorganisms as well as the infected gastric epithelial or somatic cells as source of essential nutrients [5,6] to provide the essential nutrition needed by the whole body. As the gut microbiome may be composed of 100 times more genes than that found in the entire human genome, the human microbiome is an indispensable source of metabolites for the human body.

Nevertheless, as our immune system also contributes to nutrition acquisition by degrading human microbiota, pathogens and damaged body tissue cells, over-nutrition may occur, which may cause lipotoxicity and further tissue damage [7,8], promoting chronic inflammation and fuelling microbial dysbiosis, which might result in adverse effect to our health.

Page 1, line 32, “gut diversity” should be “gut microbiota diversity”.

Page 8, line 244, “significant alter” should be “significantly alter”.

The following references may be included in the revision of the manuscript:

1.         Levin, B.R.; Baquero, F.; Ankomah, P.P.; McCall, I.C. Phagocytes, Antibiotics, and Self-Limiting Bacterial Infections. Trends Microbiol. 2017, 25, 878–892.

2.         Levin, B.R.; Antia, R. Why we don’t get sick: The within-host population dynamics of bacterial infections. Science 2001, 292, 1112–1115.

3.         Troisi, J.; Venutolo, G.; Pujolassos Tanya, M.; Delli Carri, M.; Landolfi, A.; Fasano, A. COVID-19 and the gastrointestinal tract: Source of infection or merely a target of the inflammatory process following SARS-CoV-2 infection? World J. Gastroenterol. 2021, 27, 1406–1418.

4.         Jorgensen, I.; Rayamajhi, M.; Miao, E.A. (2017) Programmed cell death as a defence against infection. Nat. Rev. Immunol., 2017, 17(3), 151-64. DOI: 10.1038/nri.2016.147.

5.         McFall-Ngai, M.; Hadfield, M.G.; Bosch, T.C.; Carey, H.V.; Domazet-Loso, T.; Douglas, A.E.; Dubilier, N.; Eberl, G.; Fukami, T.; Gilbert, S.F.; et al. Animals in a bacterial world, a new imperative for the life sciences. Proc. Natl. Acad Sci. USA 2013, 110, 3229–3236.

6.         Davari M, Moghaddam HR, Soola AH (2021) Identifying the Predictors of Self-Management Behaviors in Patients with Diabetes Based on Ecological Approach: A Systematic Review. Curr Diabetes Rev 17(6): e102620187197. DOI : 10.2174/1573399816666201026161009

7.         Saltiel, A.R.; Olefsky, J.M. Inflammatory mechanisms linking obesity and metabolic disease. J. Clin. Investig. 2017, 127, 1–4.

8.         Garbarino, J.; Sturley, S.L. Saturated with fat: New perspectives on lipotoxicity. Curr. Opin Clin. Nutr. Metab. Care 2009, 12, 110–116.

Author Response

Comment 1: When we talk about gut microbiota, we need to take the human host immunity response into account [1-3]. Thanks to the immune responses like phagocytosis, xenophagy, and programmed cell-death like pyroptosis and necroptosis [4], most of microorganism infections are self-limiting (and become the so-called gut microbiota), and our immune system will use metabolic products of these microorganisms as well as the infected gastric epithelial or somatic cells as source of essential nutrients [5,6] to provide the essential nutrition needed by the whole body. As the gut microbiome may be composed of 100 times more genes than that found in the entire human genome, the human microbiome is an indispensable source of metabolites for the human body. Nevertheless, as our immune system also contributes to nutrition acquisition by degrading human microbiota, pathogens and damaged body tissue cells, over-nutrition may occur, which may cause lipotoxicity and further tissue damage [7,8], promoting chronic inflammation and fuelling microbial dysbiosis, which might result in adverse effect to our health. The following references may be included in the revision of the manuscript.

Reply 1: Thank you for the thoughtful comments and references provided. We have incorporated them as far as possible in our revised discussion section, "An additional consideration is the immune response to probiotics by the human host. Immune responses like phagocytosis, xenophagy, and programmed cell-death like pyroptosis and necroptosis [48,49] render most invaders harmless and microbial infections self-limiting. Our immune system also contributes to nutrition acquisition by degrading human microbiota, pathogens and damaged body tissue cells as it is able to utilize some of the metabolic products from these microorganisms as well as the infected gastric epithelial or somatic cells as a source of essential nutrients [50,51]. As the gut microbiome is composed of 150 times more genes than that found in the entire human genome, the human microbiome is an indispensable source of metabolites for the human body [52,53], and these may modulate the overall effect on health and disease states."

Comment 2: Page 1, line 32, “gut diversity” should be “gut microbiota diversity”.

Reply 2: This has been corrected. Thanks for spotting the mistake.

Comment 3: Page 8, line 244, “significant alter” should be “significantly alter”.

Reply 3: This has been corrected. Thanks for spotting the mistake.

Round 2

Reviewer 1 Report

According to the comments of editors and reviewers, the overall quality of this manuscript has been improved after the author's modification. The field of the author discussed is pioneering. Indeed, there are limited studies on the application of probiotics in the treatment of Major Depressive Disorder, and more data are needed to confirm. However, the overall idea is worthy of recognition and is also worthy of reference for subsequent research on probiotics for treating MDD. I feel that it is suitable for publication in this journal.